# TRAVELLING SALESMAN PROBLEM GOES SPARSE WITH GRAPH NEURAL NETWORKS

## ABSTRACT

Machine learning based approaches to solve the Travelling Salesman Problem (TSP) have achieved astonishing performance in the last years. A large number of works proposing such approaches use a type of encoder in their underlying frameworks to learn vector representations of the given problem. Since TSP can easily be interpreted as a graph theoretic problem, Graph Neural Networks (GNNs) have been a popular encoder architecture for this task. However, most papers ignore that GNNs are not designed to operate on complete graph instances like the TSP. We therefore propose two data preprocessing methods for GNNs to make the TSP instances sparse: a nearest neighbor based heuristic and a method based on minimum spanning tree called 1-Tree. We show that making the underlying TSP instances sparse by deleting unpromising edges in the preprocessing step improves the performance of the overall learning framework while, at the same time, the runtime decreases. In particular, the proposed method achieves an up to $\times 2$ performance improvement w.r.t. the optimality gap and a decrease in runtime by $10\%$ during training and validation, when applied to GCNs. For GATs, the improvements in regards of runtime and optimality gap are even bigger when sparsifying the data first: We report up to $\times 22$ improvements for the optimality gap while reducing the runtime by $50\%$.

## 1 INTRODUCTION

In recent years, graph neural networks (GNNs; see Appendix A.1.1) have emerged as a powerful architecture when dealing with graph structured data like molecules, social networks or traffic models (Wu et al., 2020). Moreover, concurrently, many papers dealt with routing problems like the travelling salesman problem (TSP; see Appendix A.1.2) or the capacitated vehicle routing problem (CVRP) using machine learning (see Section 2). Despite the fact that many of these papers take completely different approaches and learning paradigms, most of these architectures require some sort of *encoder* in their framework which produces vector representations for the nodes in the problem instance. In this work, we show that GNNs can serve as powerful encoders in machine learning frameworks for solving the TSP if applied correctly. So far, most other papers using GNNs as their encoder architecture naively ran their GNN on the the whole *dense* TSP graph. We, on the contrary, investigate the influence of preprocessing TSP graphs by deleting edges, which are unpromising to be part of the optimal TSP solution, before passing the graphs to the GNN. This sparsification allows the GNN to focus on the relevant parts of the problem and, therefore, to produce better embeddings for further processing in the overall framework as we show in the experiments in Section 4. In particular, we provide the following contributions:

- We propose two data preprocessing methods for GNNs when operating on TSP data that makes the corresponding TSP instances sparse by deleting unpromising edges: The simple $k$-nearest neighbors heuristic and 1-Trees, a minimum spanning tree based approach which was previously used successfully in the initialization of the powerful LKH algorithm (Helsgaun, 2000). We then compare both methods in regards of their capability of keeping the optimal edges in the sparse representation of a TSP instance.

- We evaluate our sparsification methods with two different GNN architectures, namely, *Graph Attention Networks* (GAT) and *Graph Convolutional Networks* (GCN) on two different data distributions and show that performance increases in all settings when sparsifi-

cation strategies are applied. We show, for example, that the optimality gap achieved when training a GAT encoder on 20000 TSP instances of size 100 on the uniform data distribution decreases from $15.99\%$ to $0.72\%$ when sparsifying the instances correctly - a $22\times$ improvement. On the same dataset, the performance of the GCN improves by a factor of $2\times$ from a gap of $2.8\%$ to $1.4\%$. An overview of how the sparsification is incorporated in the overall learning framework is given in Fig. 3.

- We examine the sparsification level (i.e., the number of edges that should be deleted) in relationship to dataset size, GNN architecture, and data distribution. In particular, we show that GCNs favor sparser graphs (keeping only the 3 most promising edges for each node in a TSP instance of size 100), while GATs have a tendency of performing best when keeping approximately half of all edges (i.e., the 50 most promising edges for each node in TSP instances of size 100).

- Moreover, we investigate the relationship between runtime and sparsification level as the aggregation operations of GNNs result in a longer computation time when more edges are present in a graph. We achieve a reduction in training time by approximately $10\%$ for GCNs and $50\%$ for GATs, depending on the exact sparsification level. In addition, we report a reduction of up to $60\%$ in validation time for GATs and approximately $10\%$ for GCNs.

To the best of our knowledge we are the first ones to investigate the importance of graph sparsification as a form of GNN data preprocessing for the TSP. We believe that our preprocessing can, together with suitable GNNs, be adapted for many existing machine learning frameworks dealing with the TSP. We further hypothesize that it can give impact to other combinatorial optimization problems where initial conditions are of paramount importance.

## 2 RELATED WORK

Papers tackling routing problems such as the TSP or CVRP with machine learning approaches can be grouped by different characteristics. One possibility is, for example, to group them by their learning paradigm (i.e., supervised learning, reinforcement learning or even unsupervised learning). Another possibility is to group them by the underlying neural architecture, typically GNNs, RNNs or attention based models like transformers (or combinations of these architecture types). In this paper, we stick to the most common categorization: Grouping them by the way machine learning is used within the overall framework.

**Encoder-decoder approaches**: Here, an encoder architecture produces embeddings for each of the nodes in the instance, which are then autoregressively selected by the decoder architecture to form a tour. By masking invalid decisions, a feasible solution is ensured. Examples are Deudon et al. (2018); Nazari et al. (2018); Kool et al. (2019); Ma et al. (2019); Kwon et al. (2020); Xin et al. (2021a); Jin et al. (2023).

**Search-based approaches**: Here, the architecture is trained to learn a sort of cost metric which can later be transformed into a valid solution by a search algorithm. For example, Joshi et al. (2019); Fu et al. (2021); Kool et al. (2022); Min et al. (2023), learn *edge probability* heatmaps which indicate how likely an edge is part of an optimal solution. Qiu et al. (2022) learn a similar edge score heatmap (which does not directly correspond to probabilities) which can later be used to guide a search algorithm. Hudson et al. (2022) learn to predict a regret score for each edge which is afterwards used in a guided local search.

**Improvement-based approaches**: Here, approaches aim to improve a given valid tour. This is either done by learning typical operations research improvement operators like $k$-opt (where $k$ edges are deleted and $k$ other edges added) like in d O Costa et al. (2020); Lu et al. (2020); Wu et al. (2021), or by learning to select and/or optimize subproblems (i.e. for example subtours or subpaths of the current solution) like in Chen & Tian (2019); Kim et al. (2021); Li et al. (2021); Zong et al. (2022); Cheng et al. (2023).

We note that, typically, machine learning models that aim to solve routing problems like the TSP contain some sort of internal encoder architecture. This is most obvious for encoder-decoder architectures. However, search-based approaches also contain some sort of encoder architecture: In this setting, the encoder is the part of the architecture responsible for creating hidden feature vectors for all the nodes in the instance which are later used to create the edge scores. In improvement-based approaches, it is less obvious, but even these frameworks use (or could use) an internal en-

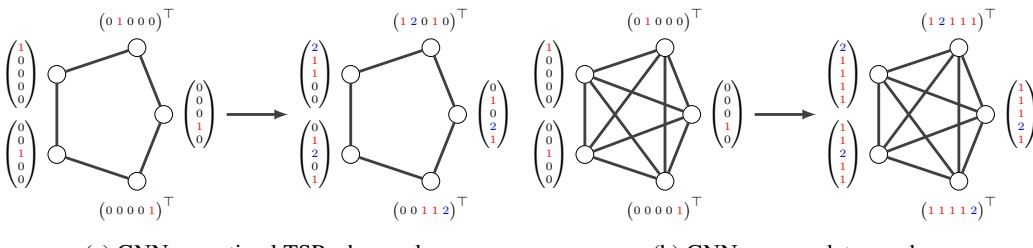

(a) GNN on optimal TSP edges only        (b) GNN on complete graph

Figure 1: Message passing - importance of sparsification

coder to produce meaningful representations (i.e., embeddings) for the nodes as it is done e.g. in d O Costa et al. (2020). We emphasize that these encoders can and have been trained with different learning techniques: e.g. Joshi et al. (2019) used supervised learning, Kool et al. (2019) used reinforcement learning and Min et al. (2023) was the first to even train in an unsupervised learning setting.

In our work, we propose a method to make GNN encoders, usable in the different learning settings mentioned above, more powerful when used to compute embeddings for the TSP in learning tasks. Therefore, our contribution is somewhat orthogonal to the above categorization as it is applicable in all settings. In particular, we point out that in e.g. Joshi et al. (2019); d O Costa et al. (2020); Fu et al. (2021); Kool et al. (2022); Min et al. (2023) GNNs were used as encoders operating on dense graph representations of the TSP. We hypothesize that these architectures would achieve better performances when using our proposed methods to sparsify the graphs first. We emphasize that Fu et al. (2021) is a paper dealing with scalability, trying to solve large TSP instances and point out that our data preprocessing is applicable in this setting as well, e.g. by sparsifying the sampled subgraphs in the overall framework (compare Fu et al. (2021) for details).

## 3    MAKING THE TSP SPARSE

### 3.1    MOTIVATION

As the TSP is a graph problem, GNNs are a natural encoder architecture choice given their success in graph related machine learning tasks. However, GNNs typically learn from the underlying graph structure to produce the node encodings (compare Morris et al. (2019); Xu et al. (2019)). As the graph in a TSP corresponds to a complete graph, there is hardly any graph structure to exploit. Moreover, in the underlying message passing operations of a GNN, information gets passed along the graph's edges. If a node is connected to irrelevant, far away nodes, its embedding will be influenced by this irrelevant information and might not represent the actual node well. Even worse, as all nodes are connected to all other nodes, they will share the exact same information in each message passing iteration, resulting in similar embeddings for all nodes. Therefore, afterwards, it will be difficult for the "decoder" part of the overall architecture to discriminate meaningfully between the different node embeddings.

We visualize this in Fig. 1. In Fig. 1a, the only edges in the graph are the ones that correspond to the actual TSP solution, whereas Fig. 1b shows a complete graph. In the figure, we assume a simple GNN that updates the feature vectors by adding the feature vectors of all nodes in the neighborhood (including itself) to the previous feature vectors. Furthermore, each node has a unique initial encoding (e.g., a one-hot encoding). We can see that in Fig. 1a, after only one update, it would be possible to decode the whole solution given the updated feature vectors: With a feature vector to start from, the next feature vector to visit should be one that has not been visited yet (easily achievable by masking) and whose entry in the current feature vector corresponds to a 1. This next node can be found by simply iterating over the other nodes' feature vectors and choosing a node whose feature vector has entry 2 at the position where the current node's feature vector has entry 1. For Fig. 1b, this would not be possible, on the other hand, as in this setting the updated feature vectors do not contain any exploitable information based on which the decoder could choose the next node to select since all entries that previously were 0 are now 1.

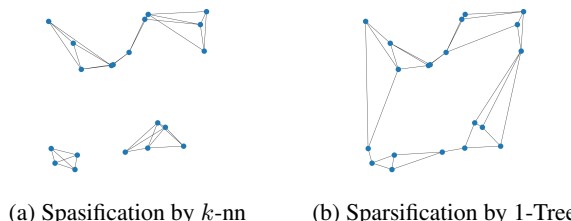

(a) Spasification by $k$-nn      (b) Sparsification by 1-Tree

Figure 2: Graph sparsified with $k$-nn and 1-Tree, keeping the 3 most promising edges for each node

Until now, most papers that used GNNs as encoders for learning routing problems completely ignored these limitations of dense TSP graphs. Only Xin et al. (2021b) and Qiu et al. (2022) tried to make the underlying graph sparse by using the $k$-nearest neighbor ($k$-nn) heuristic, where each node is only connected to its $k$ closest neighbors (using Euclidean distance). We note that the motivation of Qiu et al. (2022) to make graphs sparse was runtime related as the amount of edges in a complete graph grows quadratically in the number of nodes $n$. By using $k$-nn to make the graphs sparse, the amount of edges can be reduced from $\mathcal{O}(n^2)$ to $\mathcal{O}(kn)$. Xin et al. (2021b) acknowledges the importance of sparsification for effective training but chooses a fixed $k = 20$ and does not investigate the level of sparsification. Moreover, Xin et al. (2021b) does not propose sparsification as a general tool to create more powerful GNN encoders, but to predict edge scores and node penalties in their specific NeuroLKH framework.

## 3.2 THE SPARSIFICATION PROCESS

As was done in the few papers that tried to make TSP graphs sparse, $k$-nn is a straightforward and easy heuristic to make graphs sparse. However, this heuristic comes with a substantial theoretical drawback: The resulting sparse graph might not be connected. This is obvious if the graph's nodes are clustered. If a graph contains two far away clusters with $k + 1$ nodes each, then $k$-nn will not result in a connected sparse graph. However, even in the data distribution that is assumed in most current papers that deal with learning approaches for the TSP, a uniform and random distribution of the nodes in the unit square, unconnected graphs can occur if they are made sparse with $k$-nn (compare Fig. 2a). This problem can of course be mitigated by increasing $k$, but by this structural information gets lost and the node embeddings get flooded with unnecessary information in the message passing steps again. On the other hand, if the sparse graph is not connected, no information will flow between the different connected components in the message passing, making it difficult to learn embeddings that encode information of far away *but relevant* nodes.

In a perfect setting, the edges in the sparse graph would correspond exactly to the TSP solution (recall Fig. 1). By this, each node would exactly receive the information of its relevant neighbors. Obviously, in this case learning would not be necessary anymore as the solution is already given. Nevertheless, for a GNN to produce the best encodings possible on a sparse graph, we expect this sparse graph to:

1. Have as little edges as possible while containing all (or at least as many as possible) edges from the optimal solution

2. Be connected

Moreover, we want the sparsification to be fast. To achieve all of these goals, we propose to use the *candidate set* of the LKH algorithm (Helsgaun, 2000) as the edges in the sparse graphs. The LKH algorithm performs $k$-opt moves on a given (suboptimal) solution to improve it until convergence. $k$-opt means that $k$ edges in the current solution are substitutes by $k$ edges not present in the current solution. In order to restrict this search, the newly added edges must be included in the before mentioned *candidate set*. The LKH algorithm uses *1-Trees*, a variant of minimum spanning trees modified by a subgradient optimization to be closer to a TSP solution, to compute the candidate set (compare Helsgaun (2000) for details on how to compute the 1-Tree of a graph). Note that a 1-Tree has a cost: The sum of all edge costs in the 1-Tree.

---

**Algorithm 1:** A simple algorithm for computing the candidate set (based on Helsgaun (2000))

    **Input** : A TSP instance $V$, amount of edges to keep $k$
    **Output:** A set of candidate edges
1  $min\_oneTree = \texttt{computeOneTree}(V); \alpha = \texttt{dict}(); candidateSet = \texttt{set}()$
2  **foreach** *pair* $u, v \in V, u \neq v$ **do** // Iterate over all possible TSP edges
3      **if** $(u, v) \in min\_oneTree$ **then**
4         $\alpha\,[(u, v)] = 0$
5      **else**
6         // Compute a new 1-Tree required to contain edge $(u, v)$
7         $oneTree = \texttt{computeOneTreeForcingEdge}(V, (u, v))$
8         $\alpha\,[(u, v)] = \texttt{cost}(oneTree) - \texttt{cost}(min\_oneTree)$
9      **end if**
10 **end foreach**
11 **foreach** $u \in V$ **do** // Iterate over all TSP nodes
12    **for** $k$ *smallest* $\alpha\,[(u, v)]$ *with* $v \in V$ **do** // Choose neighbors with lowest $\alpha$
13      $candidateSet.\texttt{add}((u, v))$
14    **end for**
15 **end foreach**
16 **return** $candidateSet$

---

Given a TSP instance $V$, the candidate set is generated as outlined in Algorithm 1. The algorithm first computes $\alpha$ scores for all edges, indicating how expensive it is to enforce an edge in a 1-Tree compared to the cheapest 1-Tree possible. Afterwards, ranking by the alpha scores, the best $k$ edges for each node are kept in the candidate set. Note, that there is a much more elegant and quicker way to compute the $\alpha$ values instead of computing new 1-Trees. We refer the reader to Helsgaun (2000) for details. Motivated by the promising performance of LKH, 1-Trees seem to be an obvious choice to make TSP graphs sparse as they also meet all our requirements:

1. Only few edges for each node are required (Helsgaun (2000) states the 5 most promising edges for each node are sufficed in their test cases).

2. Spanning trees (and therefore 1-Trees) are naturally connected, leading to connected sparse graphs.

For comparison, we show the sparse versions of the graph in Fig. 2a when sparsified with the 1-Tree approach instead of $k$-nn in Fig. 2b. Note that the graph is now connected. Furthermore, we note that in our experiments (Section 3.3 and Section 4) we will make the candidate set symmetric, which means that if an edge $(u, v)$ is included, we also include $(v, u)$, as the TSP is a symmetric problem (the solution is optimal, no matter in which "direction" we travel).

In the following, we refer to the sparsification method used in the candidate set generation of the LKH as "1-Tree". "1-Tree" in combination with a certain $k$ means that the $k$ most promising edges according to the 1-Tree candidate set generation of LKH are kept for each node in the graph.

### 3.3 OPTIMAL EDGE RETENTION CAPABILITY OF SPARSIFICATION METHODS

We now investigate how good the two different sparsification methods perform in regards of keeping the optimal TSP edges in the sparse graph representation. In particular, we select 100 random graphs of size 100 of two data distributions, keep the $k$ most promising edges for $k \in \{2, 3, \ldots, 10\}$ and count for how many out of the 100 graphs all edges that are in the optimal solution of the TSP are also part of the sparse graph. The two data distributions we investigate are *uniform* and *mixed*, explained in Appendix A.2. When evaluating the candidate edges generated with 1-Tree and $k$-nn on the 100 random graphs of the different data distributions, we obtain the results in Table 4. On the uniform data distribution, for $k = 5$, 65 out of 100 sparse graphs contained all optimal edges when using 1-Tree for the sparsification. This is considerably more than for $k$-nn, where only for 5 instances all optimal edges were in the sparse graph representation, but far away from the desired 100 instances. For $k = 10$, 98 sparse graphs contain all optimal edges if the sparsification was performed with 1-Trees. On the mixed data distribution, the gap between $k$-nn and 1-Tree gets even

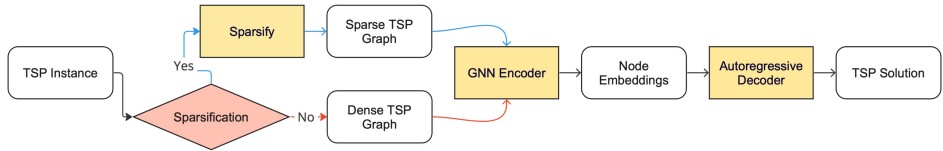

Figure 3: Flowchart presenting the overall encoder-decoder framework used in the experiments including new sparsification step (blue/top) compared to previous dense approaches (red/bottom).

bigger, which is most likely due to the coordinates of the nodes in the underlying graphs being more clustered.

Overall, even while not providing sparse graphs with 100% optimal edge coverage for small $k$ like $k = 5$, we expect 1-Tree to be nevertheless a better sparsification method than $k$-nn, especially on non-uniform, clustered data (which is more similar to real-world data) and for smaller, fixed $k$. We believe that sparsifying the graphs and, therefore, deleting the majority of non-optimal edges will allow GNN encoders operating on TSP data to focus on the relevant parts of the problem, leading to better embeddings.

## 4 EXPERIMENTS

To examine the importance of sparsification, we conduct several experiments on TSP instances of size 100. In particular, we investigate the following questions:

- How do different GNN architectures (GCNs vs GAT) perform as encoders?
- How does preprocessing the TSP instances to sparse graphs influence the performance of the GNNs compared to when using dense TSP representations?
- Which sparsification preprocessing method ($k$-nn vs 1-Tree) leads to better results?
- How do different data distributions for generating the TSP instances (*mixed* and *uniform*) influence the degree of sparsification (i.e., the choice of $k$) and sparsification method?
- How does the training dataset size $d$ (amount of TSP instances in the dataset) relate to the sparsification parameter $k$? Are attention based GNNs like the GAT able to determine the important neighbors by themselves if $d$ is big enough, making sparsification obsolete?
- How does runtime (training, validation and preprocessing times) develop for different degrees of sparsification and GNN type?

To try to answer the above questions, we train combinations of GNN architectures (GAT, GCN), data distributions (uniform, mixed), sparsification methods ($k$-nn, 1-Tree), sparsification degrees $k \in \{3, 5, 10, 20, 50\}$ and training dataset sizes $d \in \{500, 1000, 5000, 20000\}$ as encoders. For the sake of brevity, we refer to such a combination as, e.g., (GAT, uniform, 1-Tree, $k$=3, $d$=500). Moreover, we also train combinations like (GCN, mixed, dense, $d$=1000) to evaluate the performance of the GNNs on dense TSP graphs. The GNNs are incorporated in an encoder-decoder approach with an autoregressive decoder. We visualize the overall framework in Fig. 3. The exact training settings can be found in Appendix A.3.

### 4.1 RESULTS

We provide plots (Fig. 6, Fig. 7, Fig. 8 and Fig. 9), showing the optimality gap of GNN, sparsification method, data distribution and $k$ combinations after each training epoch in the appendix for $d \in \{500, 20000\}$. All optimality gaps are in respect to the optimal solution generated by the concorde solver (Applegate et al., 2006).

Futhermore, we provide tables for each data distribution - GNN type combination, indicating the best optimality gap achieved in any epoch. The tables can be interpreted as follows: For a specific $k$ and $d$, the corresponding cell contains two values. The first one was achieved by using $k$-nn as the sparsification method, the second one was achieved by using 1-Tree. Cells were 1-Tree achieves a

Table 1: Results - GAT - Optimality gap in percent

| Uniform - GAT | | | | | | |
|---|---|---|---|---|---|---|
| d \ k | 3 | 5 | 10 | 20 | 50 | dense |
| 500 | 3.12 / 3.1 | 3.95 / 3.24 | 3.32 / 3.41 | 3.52 / 3.48 | 3.46 / **3.02** | 16.01 |
| 1000 | 2.64 / **2.37** | 2.68 / 2.57 | 2.76 / 2.58 | 2.62 / 2.54 | 2.49 / 2.5 | 16.16 |
| 5000 | 1.92 / 1.43 | 1.92 / 1.7 | 1.91 / 1.69 | 1.48 / 1.39 | 1.54 / **1.37** | 16.14 |
| 20 000 | 1.07 / 0.88 | 0.98 / 0.93 | 1.01 / 0.9 | 0.89 / 0.83 | 0.81 / **0.72** | 15.99 |

| Mixed - GAT | | | | | | |
|---|---|---|---|---|---|---|
| d \ k | 3 | 5 | 10 | 20 | 50 | dense |
| 500 | 4.44 / 4.27 | 4.6 / 4.5 | 4.11 / 4.46 | 4.12 / 4.34 | **3.89** / 4.42 | 14.67 |
| 1000 | 3.7 / 3.22 | 3.44 / 3.55 | 3.47 / 3.47 | 3.29 / 3.41 | 3.13 / **3.03** | 14.48 |
| 5000 | 2.56 / 2.07 | 2.29 / 2.3 | 2.21 / 2.32 | 2.07 / 2.12 | **1.85** / 1.86 | 14.56 |
| 20 000 | 1.78 / 1.25 | 1.42 / 1.47 | 1.42 / 1.5 | 1.34 / 1.4 | 1.09 / **1.07** | 2.23 |

better performance than $k$-nn are colored red, whereas cells where $k$-nn performs better than 1-Tree are colored blue. White cells indicate that both initializations led to the same performance, or, in the case of "dense" cells, that no sparsification was performed. Furthermore, each row (corresponding to a certain dataset size $d$) has one bold and one underlined value. The bold value corresponds to the very best optimality gap in the row, the underlined value to the second best score.

**Uniform - GAT**: First, we note that the uniform distribution part of Table 1 is mostly reddish, indicating a tendency for 1-Tree to perform better. Furthermore, we note that the GNNs operating on dense graphs led to an overall performance that was much worse than any sparsified combination, no matter the dataset size. Depending on the dataset size, the gap for the best sparse instances is between $\times 5$ to $\times 22$ better than the gap achieved on the dense graph representations. We hypothesize that this is due to two reasons: First, the overall message passing architecture is not optimized for dense graphs. Secondly, there is possibly not enough training data or time for the GNN to figure out via the attention mechanism which of the neighbors are important and which are not - even in the case of 20000 training instances. We note, however, that $k = 50$ led to a very good performance overall, even on small datasets, as we can see by the abundance of underlined and bold values in the corresponding column. This is interesting, keeping in mind the poor performance of dense graphs. We note, nevertheless, that for $d = 500, 1000$ the 1-Tree based sparsification with $k = 3$ led to good results too, producing the second best and best results, respectively. This aligns with our expectations that it is more difficult for the GAT to figure out the importance of nodes in settings with more edges (i.e. higher $k$) if the dataset is rather small.

**Mixed - GAT**: Contrary to the uniform dataset, this table, the lower part of Table 1, is mostly bluish, indicating a superiority of $k$-nn over 1-Tree. This is surprising, given that the data distribution leads to clustered coordinates where $k$-nn is in a disadvantage when computing connected, sparse graph representations as discussed in Section 3.3. Possibly, this indicates that the decoder in the overall architecture can mitigate the effect disconnected sparse graphs have on the GNN. We note, furthermore, that $k = 50$ generally leads to the best results. This is again unexpected as the performance of the GNNs operating on the dense graphs was much worse again, with the optimality gaps for GNNs operating on sparse graphs being up to almost $\times 8$ times better. We hypothesize that, given the limited training data, the attention mechanism is unable to learn the importance of all neighbors in dense graphs whereas it seems to be capable to detect important neighbors if $k = 50$. We note, however, that for $d = 20000$ something interesting happened: While the gap for the dense GNN encoders was around $14\%$ for the smaller datasets, it was only $2.23\%$ on the largest dataset. By examining the plot of the learning curve for $d = 20000$ in Fig. 9, we can see that the model had an unexpected performance gain after approximately 800 epochs. Apparently, in this particular setting, the GAT was able for the first time to figure out the importance of neighboring nodes by itself - using the attention mechanism. It is open whether the performance of the model would be able to catch up to - or even surpass - the models operating on sparsified data if training was performed for even longer. Note that we further investigated this phenomenon in Appendix A.3.1.

**Uniform - GCN**: The upper part of Table 2 is predominantly red again, except for $k = 5$ and some

Table 2: Results - GCN - Optimality gap in percent

| Uniform - GCN | | | | | | |
|---|---|---|---|---|---|---|
| d \ k | 3 | 5 | 10 | 20 | 50 | dense |
| 500 | 3.33 / **3.01** | 3.65 / 4.06 | 4.24 / 4.46 | 4.44 / 4.49 | 4.58 / 4.32 | 4.6 |
| 1000 | 2.97 / **2.26** | 3.26 / 3.62 | 3.87 / 3.82 | 3.93 / 3.92 | 4.1 / 3.92 | 4.11 |
| 5000 | 2.36 / **1.7** | 2.55 / 2.81 | 2.95 / 2.92 | 3.19 / 3.02 | 3.29 / 3.15 | 3.28 |
| 20 000 | 1.91 / **1.41** | 2.04 / 2.35 | 2.59 / 2.44 | 2.7 / 2.49 | 2.8 / 2.58 | 2.8 |

| Mixed - GCN | | | | | | |
|---|---|---|---|---|---|---|
| d \ k | 3 | 5 | 10 | 20 | 50 | dense |
| 500 | 4.47 / **3.29** | 4.58 / 4.76 | 4.8 / 4.98 | 5.04 / 5.1 | 5.48 / 5.18 | 5.21 |
| 1000 | 3.87 / **2.7** | 4.03 / 4.12 | 4.31 / 4.33 | 4.5 / 4.36 | 4.88 / 4.66 | 4.72 |
| 5000 | 3.11 / **1.83** | 3.14 / 3.14 | 3.46 / 3.41 | 3.66 / 3.47 | 3.93 / 3.7 | 3.74 |
| 20 000 | 2.62 / **1.53** | 2.72 / 2.67 | 2.97 / 2.87 | 3.19 / 2.95 | 3.36 / 3.17 | 3.21 |

entries for $d = 500$. More importantly, the best performance in this setting was achieved by $k = 3$ for all dataset sizes. Moreover, we note that for the GCN, the dense graphs led to a rather reasonable overall performance, compared to the performance of dense GAT encoders. Nevertheless, for $d = 500$, the optimality gap improves from $4.6\%$ (dense TSP graphs) to $3.01\%$ (sparsified with 1-Tree with $k = 3$). This corresponds to a $\times 1.5$ performance increase. For $d = 20000$ the optimality gap improvement is even bigger, from $2.8\%$ for dense TSP graphs to $1.41\%$ for sparse graph representations generated by 1-Tree with $k = 3$, a $\times 2$ improvement.

**Mixed - GCN**: In this setting, the results (visible in the lower part of Table 2) are similar to the results produced by the GCN on the uniform distribution. The table is overall reddish again, indicating that 1-Tree performed often better than $k$-nn. The only exceptions are for combinations with smaller $d$ and intermediate $k$. The best results were achieved by $k = 3$ again for all dataset sizes. The optimality gap improvements are also similar to the uniform data distribution case, with around $\times 1.5$ improvement for $d = 500$ and a $\times 2$ improvement for $d = 20000$ (comparing dense graph representations to the best sparse ones).

**Summary**: We summarize that for the GCN, smaller $k$ led to the overall best results, whereas for the GAT there is a tendency for bigger $k$ (but not dense graphs!) to lead to the best results. We note that dense GAT encoders performed unexpectedly bad, keeping in mind that it has been used successfully in other works like Kool et al. (2019). We hypothesize that this is because all of our models are implemented in PyTorch Geometric Fey & Lenssen (2019) with actual message passing operations, whereas previous papers relied on custom designed "GNNs" implemented in plain PyTorch Paszke et al. (2017). We note, however, that with enough training our GATs can also learn in the dense setting as has happened on the mixed dataset with $d = 20000$. We further note that on the uniform data distribution and $d \leq 1000$ the best GCN encoders performed better than the best GAT encoders. On the mixed data distribution, GCNs performed better than GATs for $d \leq 5000$. To conclude, we note that sparsification does increase the overall performance. For extensive sparsification, i.e., for $k = 3$, 1-Tree based data preprocessing always performs better than $k$-nn. For $k = 50$ on the other hand, it is more arbitrary whether $k$-nn or 1-Tree based sparsification leads to better results. This is expected, however, as the overlap between the two sparse graphs is most likely becoming bigger as $k$ grows, making the two sparsification methods more similar.

## 4.2 RUNTIMES

The runtimes are hardly influenced by different data distributions and initialization types, meaning a (GAT, uniform, 1-Tree, $k = 10$, $d = 5000$) will have approximately the same runtime as (GAT, mixed, $k$-nn, $k = 10$, $d = 5000$) in terms of training and validation time. What does have a significant influence on the runtime are the choosen $k$ value (as the aggregation operations in the GNN are much faster for smaller $k$ as the aggregated neighborhoods are smaller), the dataset size $d$ and the chosen architecture GCN vs GAT. The different runtimes of the architectures stem mostly from the different amount of layers (3 vs 6 layers respectively) but also the additional parameters

Table 3: Training time GAT in hours

| k
d | 3 | 5 | 10 | 20 | 50 | dense |
|---|---|---|---|---|---|---|
| 500 | 1.11 | 1.11 | 1.12 | 1.39 | 1.94 | 2.5 |
| 1000 | 2.22 | 2.23 | 2.5 | 2.81 | 3.88 | 5.1 |
| 5000 | 11.37 | 11.76 | 12.81 | 14.61 | 19.83 | 25.93 |
| 20 000 | 46.3 | 46.88 | 50.68 | 58.46 | 79.55 | 103.4 |

the GAT has to learn for the attention scores. As a runtime representative, we chose the runtimes on the uniform dataset when using 1-Tree as the sparsification method. The training times of the GAT in hours can be found in Table 3. The times for the GCN are in the appendix in table Table 5. We observe that the training time of the GAT is approximately $\times 2$ as long when using dense TSP graphs compared to very sparse TSP graphs with $k = 3$. For the GCN, the runtime improvement is approximately $10\%$ when using sparse graphs.

The validation times for both GNN architectures can be found in the appendix in Table 6. We note that the time the GAT needs to validate 1000 instances decreases from almost 60 seconds to 20 seconds, when using sparse graphs with $k = 3$ compared to dense graphs. This is a $\times 3$ improvement. For the GCN, the difference is less pronounced again, with a decrease in validation time from 21 seconds to 18 seconds, approximately a $10\%$ improvement. While the runtime advantage during training for sparse TSP instances is obvious, it is only several seconds at validation time (even if, in the case of the GAT, this is till a $\times 3$ improvement). In order to determine whether sparsification really leads to a runtime improvement at inference time, we need to take into account the time spent for sparsification as well. Therefore, we report the preprocessing times in Table 7. Note that in this case, we have to separate the times for 1-Tree and $k$-nn, as the 1-Tree computations take more time than applying a simple $k$-nn heuristic. We further have to differentiate between datasets used by the GCN and the GAT, as the GCN requires the edge weights $e_{i,j}$ which are also computed in the data preprocessing. Moreover, we observe that dense graphs have a preprocessing times as well, as in this case we still have to generate PyTorch Geometric data objects and, in the case of the GCN, generate the edge weights. Note that the preprocessing times are reported for 10000 instances, while the validation dataset is only 1000 instances big. Comparing Table 6 and Table 7, we can see that even if we add $1/10$th of the 1-Tree preprocessing times to validation times, the GAT is still always faster when operating on the sparse graphs. For the GCN, the overall runtime at inference time is also better for all sparsified combinations, except for the case with 1-Tree with $k = 50$. However, recalling Section 4.1, the best sparsification degree for the GCN was $k = 3$, which has a better overall sparsification and validation time than the times achieved in the dense settings.

## 5 CONCLUSION

In this work, we proposed two data preprocessing methods for GNN encoders when learning to solve the travelling salesman problem: $k$-nearest neighbors and a 1-Tree based approach. The aim of both methods is to delete unnecessary edges in the TSP graph making it sparse and allowing the GNN to focus on the relevant parts of the problem. We analyse both sparsification methods from a theoretical point of view, pointing out that TSP instances sparsified with 1-Tree are always connected. Moreover, we validate on random instances that the 1-Tree based approach is less likely to delete optimal edges of the TSP tour when producing the sparse graph. We show that graph neural networks can increase their performance when sparsification is performed in a data preprocessing step. At the same time, GNNs are able to improve their overall runtime, when operating on these sparse TSP graph representations. Our experiments show that the exact sparsification degree (i.e. the amount of edges to be kept in the sparse graph) is dependent on the data distribution and the dataset size, as well as the chosen GNN architecture.

We emphasize that our GNN encoder is independent of the chosen learning paradigm. Moreover, we note that our proposed data preprocessing for GNN encoders is highly flexible and can be incorporated in many learning based frameworks to solve the TSP. We leave it open to feature work to design search-based or improvement-based approaches, as well as approaches dealing with scalability, using GNNs as encoders with our proposed data preprocessing.

ACKNOWLEDGMENTS

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

# A APPENDIX

## A.1 PRELIMINARIES

### A.1.1 GRAPH NEURAL NETWORKS

Graph Neural Networks (GNNs) are a class of neural architectures that allows for capturing the underlying graph structure of its input. A GNN computes feature vectors for all the nodes in the underlying input graph. In each layer, a node $i$ receives information from all other nodes $j$ it is connected to via an edge (called *neighborhood $N(i)$*). This process is called *message passing*. A node's feature vector is updated by aggregating the set of received messages. The aggregated information is typically normalized, passed through a neural layer and added to the old node vector representation. Additional activation functions applied to the intermediate outputs lead to non-linearities and the capability to learn complex functions.

In this work, we focus on two different GNN types: A version of the Graph Convolutional Network (GCN) (originally introduced by Kipf & Welling (2017)) which was adapted by Morris et al. (2019):

$$x_i^\ell = \mathbf{W_1}^\ell \cdot x_i^{\ell-1} + \mathbf{W_2}^\ell \cdot \sum_{j \in N(i)} e_{j,i} \cdot x_j^{\ell-1}$$

Here $x_i^\ell$ is the feature vector of node $i$ in the $\ell$th layer, $e_{j,i}$ is a normalization score, indicating how important the feature vector of neighboring node $j$ is to node $i$ and $\mathbf{W_1}^\ell, \mathbf{W_2}^\ell$ are learnable weight matrices.

Moreover, we use the version of the Graph Attention Network (GAT; originally by Veličković et al. (2018)) proposed by Brody et al. (2022), with the main difference compared to the GCN being that this architecture does not have fixed scores $e_{j,i}$ which are passed to the network, but these scores are learned by the GNN itself by using the attention mechanism (Vaswani et al., 2017). Moreover, this GNN also allows us to use edge features (information for each edge in the graph) which can be used for the learning.

### A.1.2 TRAVELLING SALESMAN PROBLEM

The Travelling Salesman Problem (TSP) is a combinatorial graph problem. The problem consists of a set of cities (formally nodes or vertices) $V = \{1, 2, \ldots, n\}$ and a distance or cost function $c : V \times V \mapsto \mathbb{R}$, which can, but does not need to be, symmetric. The goal is to find a tour that visits every node exactly once and ends in the same city where it started (i.e. a Hamiltonian cycle) while minimizing the travelled distance. Typically, it is possible to travel from any city to any other city. Therefore, from a graph perspective, the problem then corresponds to a complete graph (without self-loops) $G = (V, E)$ where $V$ as above and $E = \{(v, u)|u, v \in V, u \neq v\}$. In our setting, we assume a symmetric cost function which corresponds to the Euclidean distance between the nodes, given each node has a pair of coordinates $(x, y)$.

## A.2 OPTIMAL EDGE RETENTION CAPABILITY - DATA DISTRIBUTIONS AND TABLES

As discussed in Section 3.3, we use two different data distributions to test the ability of our sparsification methods to retain optimal TSP edges in the sparse graphs: *uniform* and *mixed*. So far,

the uniform data distribution has been used in most papers, tackling routing problems like the TSP or CVRP with machine learning. In this data distribution, all coordinates $(x, y)$ of the nodes are sampled uniformly at random within the unit square, i.e. $x, y \in [0, 1]$.

In the mixed data distribution (find examples in Fig. 5), we start with a uniform node distribution and afterwards apply 100 random mutation operators. The mutation operators that can be chosen from are *explosion*, *implosion*, *cluster*, *expansion*, *compression*, *linear projection* and *grid*, presented in more detail in Bossek et al. (2019). The coordinates sampled from the mixed distribution are much more clustered than the coordinates sampled from the uniform distribution, which is more similar to real world data.

Table 4 shows how our proposed sparsifcation methods, $k$-nn and 1-Tree, perform in regards of keeping all optimal TSP edges in sparse graphs for different levels of sparsification. We note that we can adapt the hyperparameters ("ASCENT_CANDIDATES" and "INITIAL_PERIOD" in Helsgaun (2000)) of the subgradient optimization in the 1-Tree generation slightly, to achieve sparse graphs with all optimal edges for all 100 graphs for $k = 10$ at the cost of approximately double the runtime for computing the 1-Trees. In particular, we can achieve this by doubling the the "ASCENT_CANDIDATES" hyperparameter and setting "INITIAL_PERIOD" to 300. We refer the reader to Helsgaun (2000) for a detailed explanation of the influence of these hyperparameters.

Table 4: Amount of graphs (out of 100) where the sparse graphs contained all optimal edges. The 65 in column "1-Tree", row "5" in the left table means that if the graphs are made sparse using 1-Trees and keeping the 5 most promising edges for each node in the graph, then for 65 graphs all edges of the optimal TSP solution were in the sparse graph, for 35 graphs there was at least one optimal edge missing.

| | Uniform | | | | Mixed | |
|---|---|---|---|---|---|---|
| k | NN | 1-Tree | | k | NN | 1-Tree |
| 2 | 0 | 0 | | 2 | 0 | 3 |
| 3 | 0 | 4 | | 3 | 0 | 14 |
| 4 | 0 | 32 | | 4 | 0 | 48 |
| 5 | 5 | 65 | | 5 | 2 | 76 |
| 6 | 23 | 81 | | 6 | 12 | 90 |
| 7 | 48 | 91 | | 7 | 21 | 94 |
| 8 | 62 | 93 | | 8 | 38 | 95 |
| 9 | 73 | 97 | | 9 | 44 | 97 |
| 10 | 85 | 98 | | 10 | 49 | 98 |

## A.3 EXPERIMENTS - SETUP

Each combination of GNN type, data distribtuion, sparsification method, $k$ and dataset size $d$ is trained for 1000 epochs on an NVIDIA T4 GPU with 16GB of VRAM and we report the performance of the architecture on the validation set after each epoch. The validation set is only used to report the current performance, it is completely independent of the training set and is also *not* used to finetune any model parameters.

For a given data distribution, the same *unprocesssed* validation set of 1000 instances was used. This means, that e.g. (GAT, uniform, 1-Tree, $k$=10, $d$=20 000) had the same unprocessed validation dataset as (GCN, uniform, $k$-nn, $k$=50, d=1000), the validation sets only differed after the preprocessing. As a result, the reported optimality gaps in the results section (Section 4.1) for different combinations are comparable, as long as the stem from the same data distribution, as the optimal solutions on the validation sets are the same.

We note that the GAT consists of 6 layers and the GCN consists of 3 layers. Moreover, the GAT uses additional edge feature vectors whose initialization are an encoding of the edge distances. We update the edge feature vectors in each layer by adding the current layers' hidden feature vectors of the edges and nodes, i.e. the feature vector of an edge $(i, j)$ in layer $\ell$ denoted as $e_{i,j}^{\ell}$ is updated as $e_{i,j}^{\ell+1} = e_{i,j}^{\ell} + x_i^{\ell+1} + x_j^{\ell+1}$, where $x_i^{\ell+1} + x_j^{\ell+1}$ are the feature vectors of node $i, j$ (the edge's endpoints) in layer $\ell + 1$.

The edge weights $e_{j,i}$ in the GCN (see Appendix A.1.1 for details) are computed in dependence of the initialization. For 1-Tree based initializations, the edges $\alpha$ scores were used, for dense and $k$-nn based initializations the edge distances were used. For a node $i$, the used $\alpha$ or distance edge scores $(j, i)$ were first normalized to the interval $[0, 1]$. Afterwards, each normalized score $n_{j,i}$ was flipped by computing $1 - n_{j,i}$. This was done in order to assign high scores to edges that previously has a low score and vice versa (a low distance or $\alpha$ score indicates a promising edge, which is why we want the corresponding $e_{j,i}$ score to be high). Subsequently, we applied softmax to the resulting flipped scores, as we want low scores to remain low but not zero (as in this case, the score $e_{j,i} = 0$ would prevent information flow from node $j$ to $i$ completely). Moreover, by this, the edge scores $e_{j,i}$ for a node $i$ sum up to 1. We show an example of the overall procedure in Algorithm 2 for a node with 4 neighbors in the (sparsified) graph.

Our code is based on the publicly available code of Jin et al. (2023). We only substituted the transformer encoder with our GNNs and adapted the datasets to represent the sparse graphs with the corresponding preprocessing. We did not change the decoder (i.e., nodes get autoregressively selected to form a valid solution) and learning is done by the original POMO inspired (see Kwon et al. (2020)) robust reinforcement learning setting. As the decoder stays the same, it is possible to choose any node after any other node when decoding, no matter if the corresponding edge was part of the sparse graph or not. This is necessary in order to always be able to find a Hamiltonian cycle in the graph which would not be possible on (e.g. but not restricted to) unconnected sparse graphs, compare Fig. 2a. An overview of our framwork including the sparsification process can be found in Fig. 3.

We note that we merely chose the architecture of Jin et al. (2023) because the used encoder-decoder architecture can be trained end-to-end and does not require any additional search algorithms for the solution generation at inference time. Moreover, the used encoder-decoder approach allowed for easy substitution of the original encoder with our GNNs operating on sparse graphs without further adjustments to the overall framework. We emphasize that we also could have chosen a search-based or an improvement-based approach to test our proposed data preprocessing for GNN encoders, where the node embeddings generated by the GNN operating on the preprocessed sparse TSP instance could be used to generate edge probability heatmaps or to determine improvement operators.

Our code will be publicly available once the paper is accepted.

---

**Algorithm 2:** Edge weight score computation - example

    **Input** : node $i$, with neighbor distances or $\alpha$ scores $scores = [0.1, 0.1, 0.2, 0.3]$
    **Output:** edge weights $e_{j,i}$ for node $i$ in the GCN

1  // Normalize scores to 0-1 range
2  $normalized = \texttt{zeroOneNormalize}(scores)$ // $normalized = [0, 0, 0.5, 1]$
3  $flipped = \texttt{list}()$
4  // Flip all scores so close neighbors have high scores
5  **foreach** $n \in normalized$ **do**
6     |  $flipped.append(1 - n)$ // $flipped = [1, 1, 0.5, 0]$
7  **end foreach**
8  // Ensure far away neighbors have non-zero score
9  $weights = \texttt{softmax}(flipped)$ // $weights = [0.3362, 0.3362, 0.2039, 0.1237]$
10  **return** $weights$

---

### A.3.1 DENSE MIXED GAT EXPERIMENT

As discussed in Section 4.1, the GAT was able to increase its performance on the dense TSP graph encodings after approximately 800 epochs on the mixed dataset. This was surprising as no other dense GAT combination was able to achieve optimality gaps under 10%. In order to investigate if the late learning after about 800 epochs was a coincidence ("bad luck", compared to the other initializations), we repeated the training of a GAT encoder on the mixed dataset with $d=20000$ for a total of 5 times. The results can be found in Fig. 4. We observe that each of the 5 training rollouts learned slightly differently. 4 out of 5 rollouts started learning considerably after about 800 - 950 epochs, 1 rollout did not have this increase in performance, producing similar results as the GAT for smaller $d$ or on the uniform data distribution. Furthermore, we observe that none of the rollouts was able to increase its performance noticeably in earlier epochs ($\ll 500$). Therefore, we hypothesize that the attention mechanism needs a certain amount of time (or data) to determine the important neighbors in the graphs by itself. We note that we chose the best rollout (i.e. rollout 5 in Fig. 4) for reporting the performance of the dense GAT on the mixed dataset part of Table 1 and in Fig. 9 for $d=20000$.

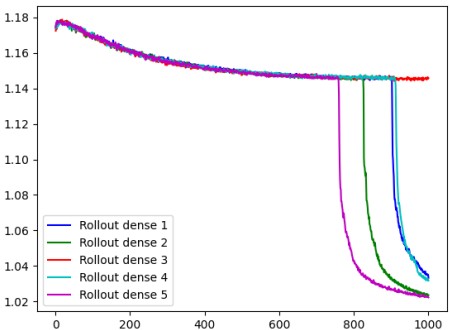

Figure 4: Performance of GAT on mixed dataset, dense graphs, $d = 20000$, several training rollouts for 1000 epochs, optimality gap (y-axis) after each epoch (x-axis)

A.4   EXPERIMENTS - RUNTIME TABLES

The following tables contain information about the training time of the GCN in hours on sparse and dense TSP instances Table 5. Moreover, Table 6 contains the time required for the different GNN architectures to evaluate 1000 sparsified or dense TSP instances at inference time. Additionally to the validation times, we provide the times required for preprocessing 10000 instances with different (or no) sparsification methods for different GNN architectures in Table 7. As described in Section 4.2, preprocessing for the GCN takes additional time to compute the edge weights. Note that, when comparing Table 6 and Table 7 to obtain the overall time required to process a sparse or dense TSP instance, the different tables contain times for a different amount of instances. The overall time can be obtained by adding $1/10$th of the times of Table 7 to Table 6.

We note that all preprocessing times were achieved on a MacBook Air with M1 CPU. The reported training and validation times stem from computations performed on an NVIDIA T4 GPU.

Table 5: Training time GCN in hours

| d \ k | 3 | 5 | 10 | 20 | 50 | dense |
|---|---|---|---|---|---|---|
| 500 | 1.0 | 0.89 | 0.99 | 0.98 | 1.11 | 1.11 |
| 1000 | 1.98 | 1.98 | 2.05 | 2.17 | 2.22 | 2.24 |
| 5000 | 10.98 | 10.69 | 11.4 | 11.18 | 11.37 | 11.69 |
| 20 000 | 43.73 | 43.2 | 43.9 | 44.23 | 45.98 | 47.84 |

Table 6: Validation times in seconds (1000 instances)

| GNN \ k | 3 | 5 | 10 | 20 | 50 | dense |
|---|---|---|---|---|---|---|
| GAT | 20.0 | 21.0 | 23.0 | 28.0 | 43.13 | 59.0 |
| GCN | 18.0 | 18.0 | 18.0 | 19.0 | 20.0 | 21.0 |

Table 7: Preprocessing times in seconds (10 000 instances)

| Comb. \ k | 3 | 5 | 10 | 20 | 50 | dense |
|---|---|---|---|---|---|---|
| GAT - $k$-nn | 19.83 | 21.0 | 23.27 | 26.8 | 44.53 | 29.23 |
| GAT - 1-Tree | 36.75 | 39.52 | 43.13 | 50.1 | 73.16 | 29.23 |
| GCN - $k$-nn | 27.86 | 29.09 | 32.19 | 40.78 | 62.43 | 56.32 |
| GCN - 1-Tree | 49.73 | 51.62 | 55.42 | 69.56 | 98.35 | 56.32 |

## A.5 MIXED DATA DISTRIBUTION - FIGURES

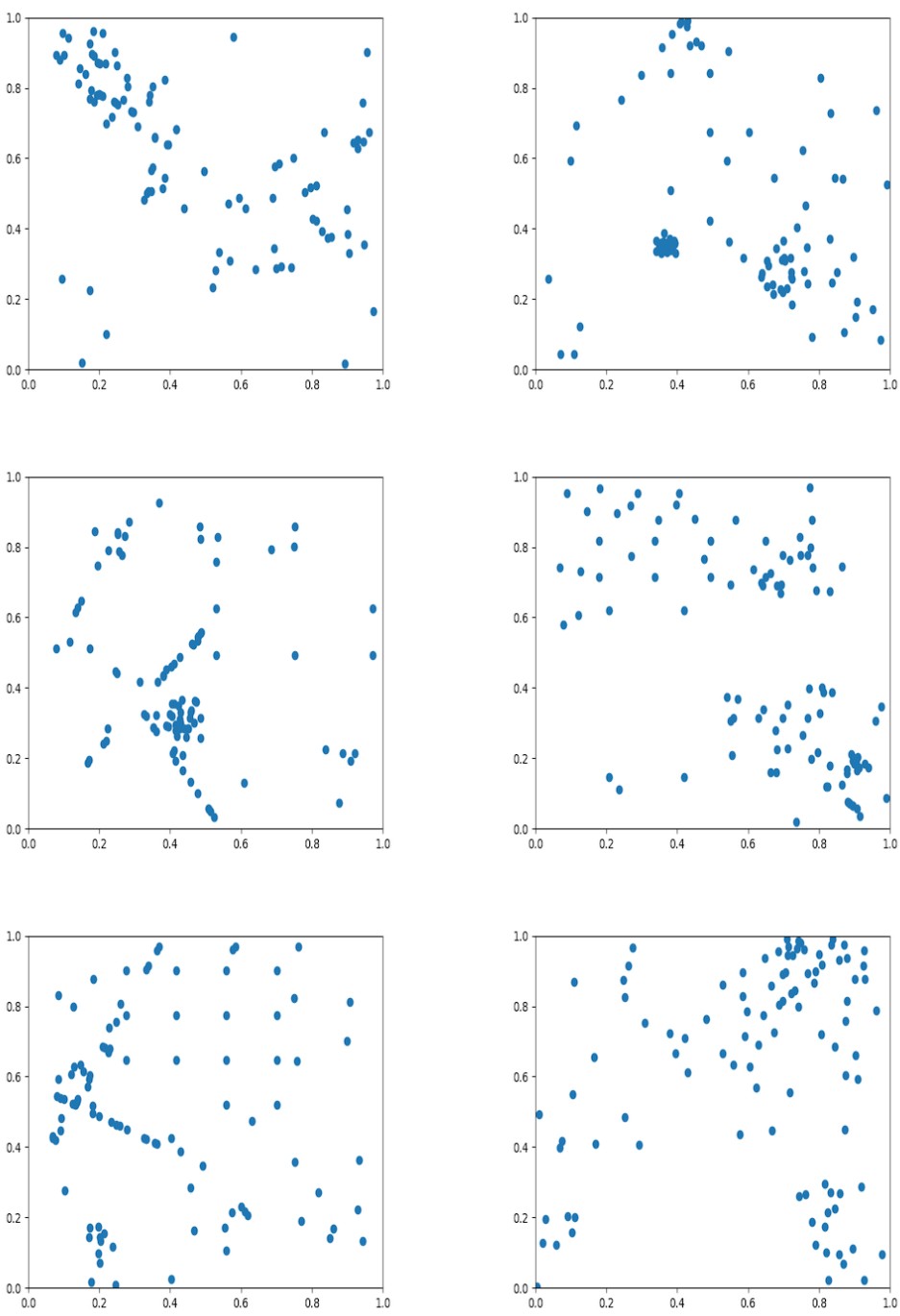

Figure 5: Visualization of 6 instances (of size 100) of the *mixed* data distribution

A.6    EXPERIMENTS - PERFORMANCE FIGURES

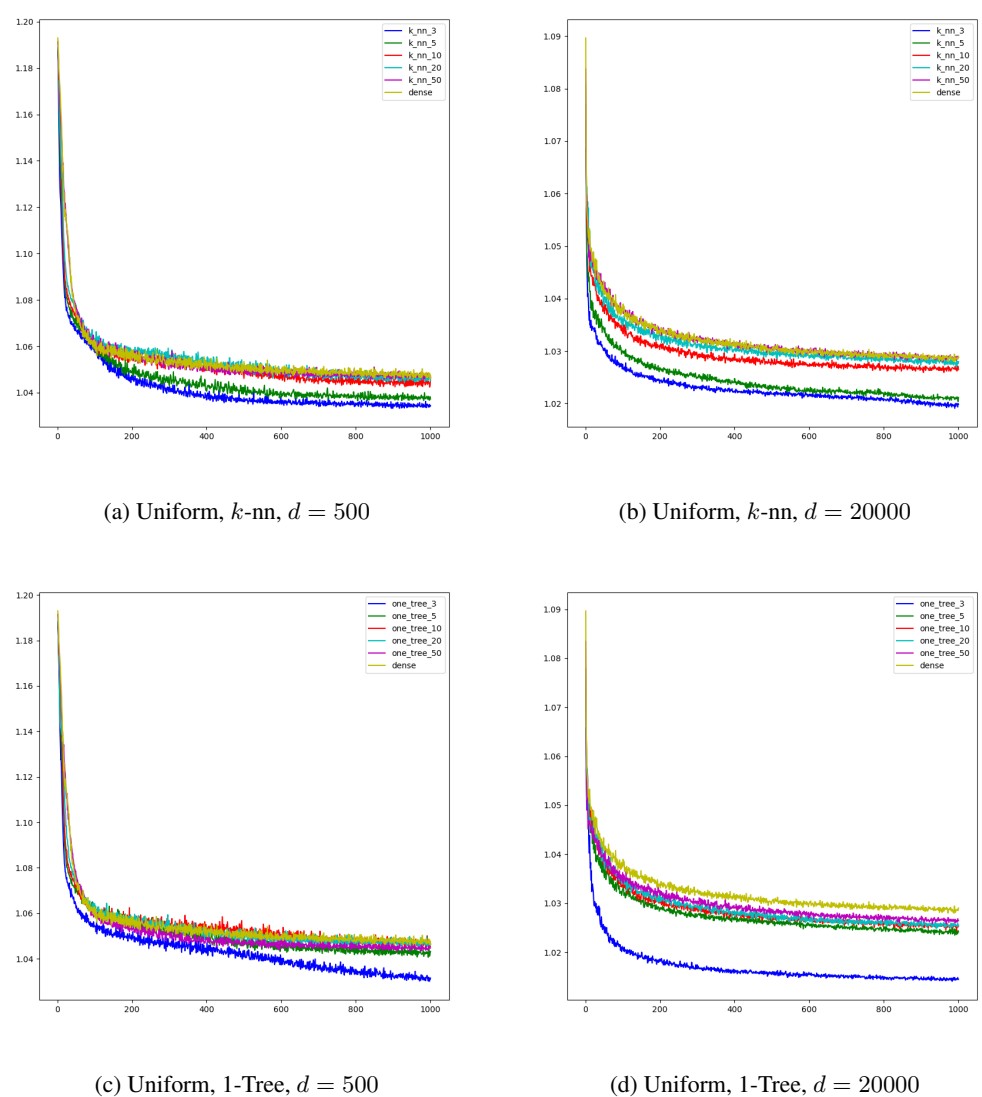

(a) Uniform, $k$-nn, $d = 500$         (b) Uniform, $k$-nn, $d = 20000$

(c) Uniform, 1-Tree, $d = 500$         (d) Uniform, 1-Tree, $d = 20000$

Figure 6: Performance of the GCN on the uniform data distribution, training for 1000 epochs, optimality gap (y-axis) after each epoch (x-axis)

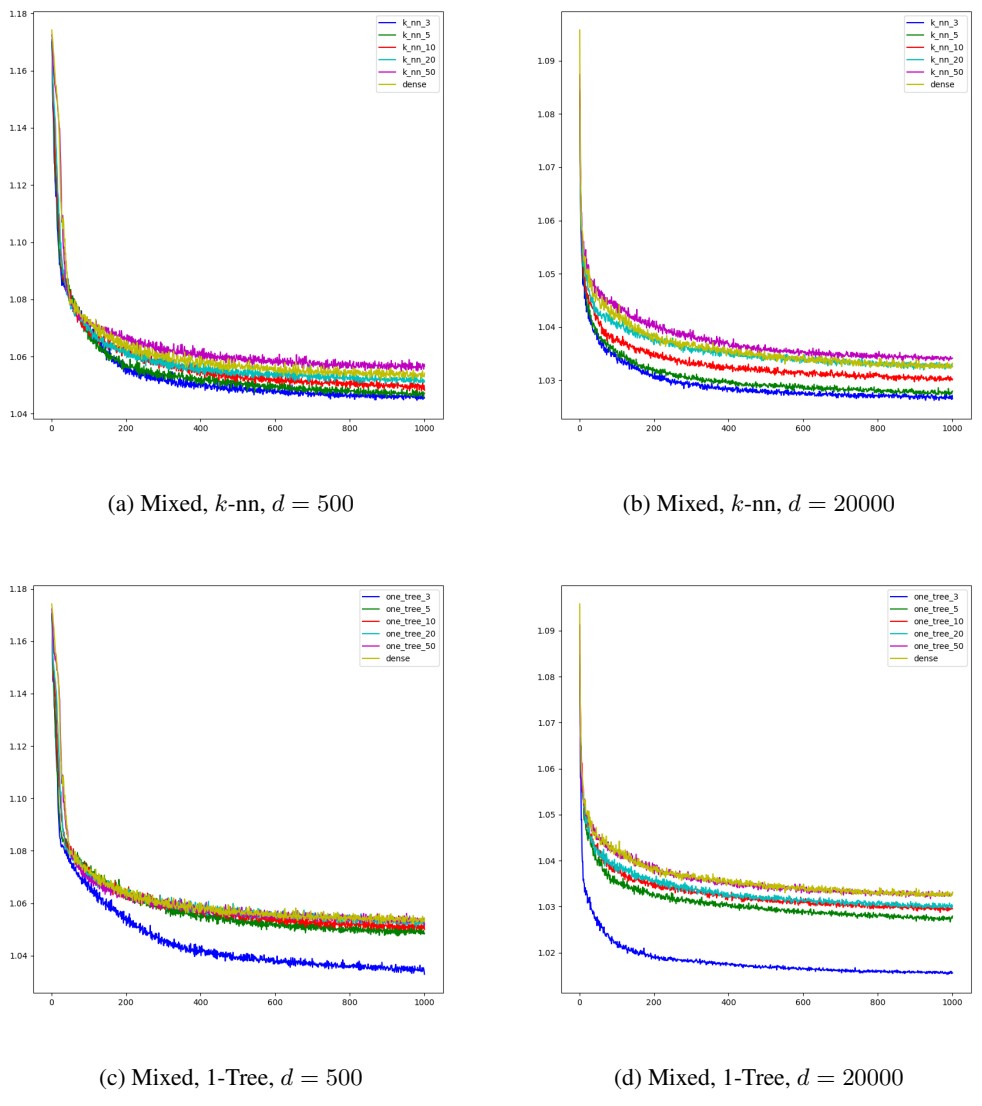

(a) Mixed, $k$-nn, $d = 500$

(b) Mixed, $k$-nn, $d = 20000$

(c) Mixed, 1-Tree, $d = 500$

(d) Mixed, 1-Tree, $d = 20000$

Figure 7: Performance of the GCN on the mixed data distribution, training for 1000 epochs, optimality gap (y-axis) after each epoch (x-axis)

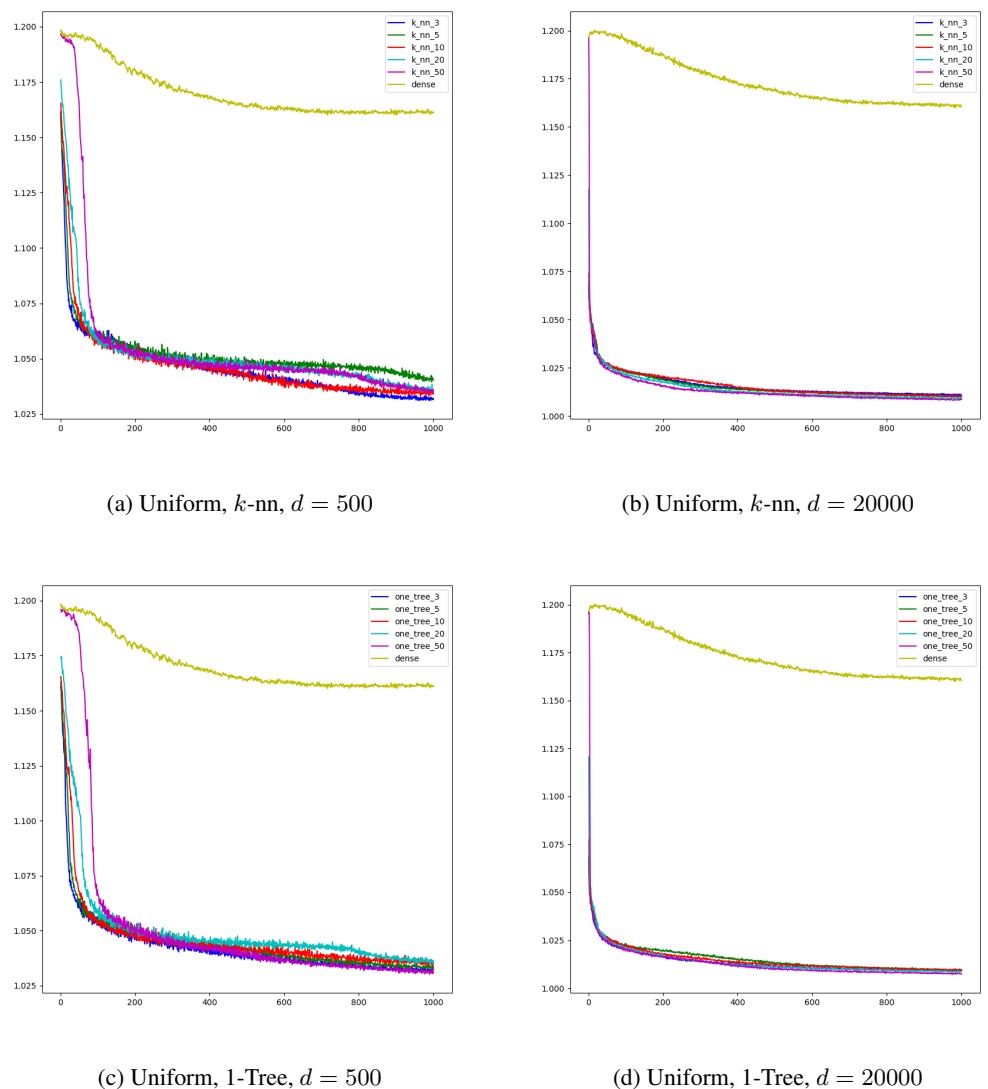

(a) Uniform, $k$-nn, $d = 500$

(b) Uniform, $k$-nn, $d = 20000$

(c) Uniform, 1-Tree, $d = 500$

(d) Uniform, 1-Tree, $d = 20000$

Figure 8: Performance of the GAT on the uniform data distribution, training for 1000 epochs, optimality gap (y-axis) after each epoch (x-axis)

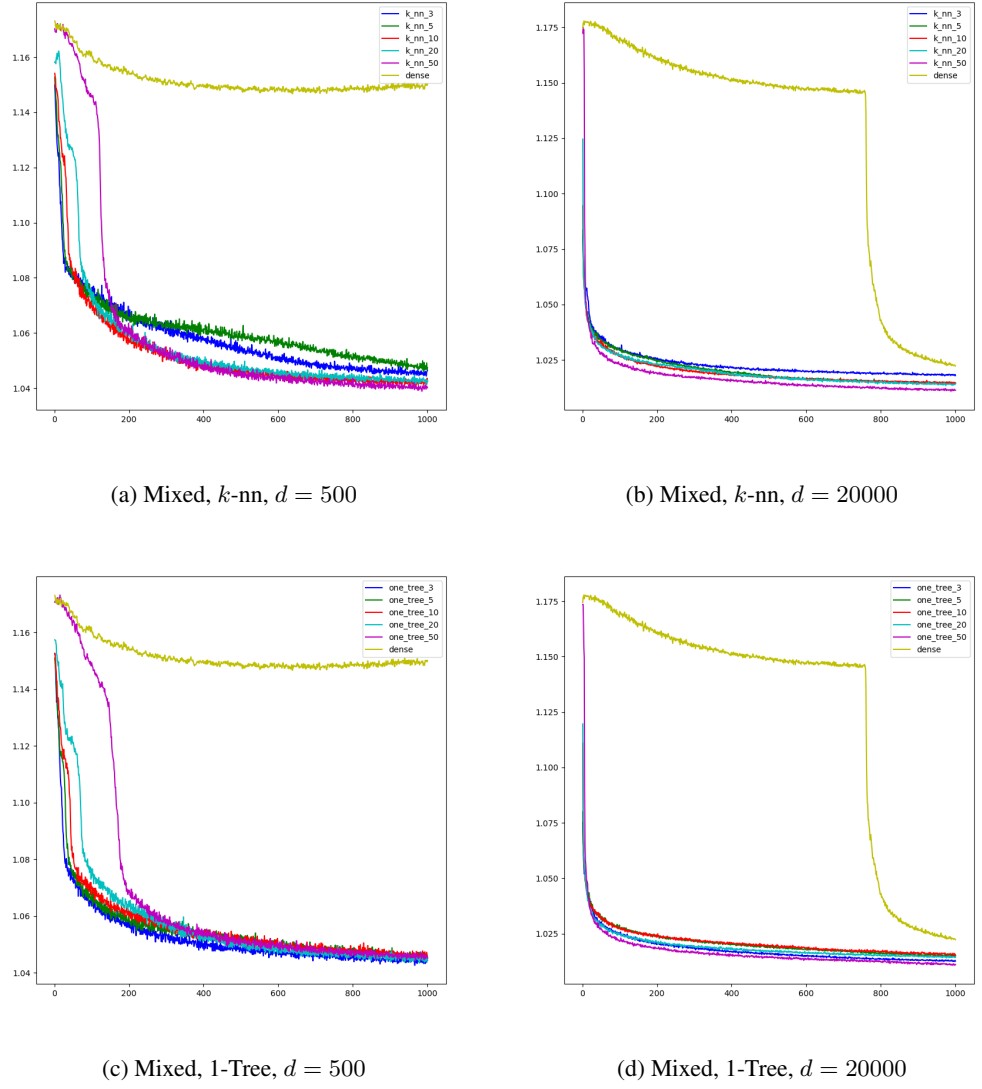

(a) Mixed, $k$-nn, $d = 500$

(b) Mixed, $k$-nn, $d = 20000$

(c) Mixed, 1-Tree, $d = 500$

(d) Mixed, 1-Tree, $d = 20000$

Figure 9: Performance of the GAT on the mixed data distribution, training for 1000 epochs, optimality gap (y-axis) after each epoch (x-axis)

