# OpenReview forum: "Travelling Salesman Problem Goes Sparse With Graph Neural Networks"
_ICLR.cc/2024/Conference — ICLR 2024 Conference Withdrawn Submission_

### Official Review · Reviewer_stG3 · 2023-10-22

**Soundness:** 3 good
**Presentation:** 2 fair
**Contribution:** 1 poor
**Rating:** 3
**Confidence:** 4

**Summary:**

This paper proposes to adopt graph sparsification techniques in the data preprocessing step for the GNN-based TSP solvers. Experiments demonstrate its effectiveness in both aspects of improving the quality of solutions and the running efficiency.

**Strengths:**

- The methods are straight-forward and easy to follow.

**Weaknesses:**

- Important baselines are missing. The experiments are more like ablation studies: The authors investigate the performance of GAT/GCN-based solvers with or without the proposed graph sparsification techniques, but does not compare the performance with other solvers mentioned in the related works. To make the contributions strong enough and the results convincing, the paper should compare with the latest methods and outperform them.

- The examples in Figure 1 do not make sense. Message passing in GNN does not propagate the features directly, but with a projection matrix (e.g. GraphSAGE). Furthermore, the problem of over-smoothing of GNN not only exist in complete graphs but also in general graphs [1]. How the proposed graph sparsification technique relieve the problem should be more clearly discussed.

- The covered problems only include the 2D tsp, which limits the the contributions of the proposed techniques.

- It lacks necessary theoretical analysis.

- The proposed 1-tree sparsification method is derivated from LKH which is a very strong TSP solver. Then the use of the technique in data preprocessing indeed brings prior knowledge to the neural solver. It is very hard to say that whether the better performance comes from the graph sparsification, or comes from the prior knowledge for TSP solving.

[1] A SURVEY ON OVERSMOOTHING IN GRAPH NEURAL NETWORKS. https://arxiv.org/pdf/2303.10993.pdf

 Based upon the above points, I believe that the work is still somehow preliminary and the paper does not meet the bar of iclr.

**Questions:**

- The size of instances is not given.
- The others are in the weakness.

---

> ### Author Response · Authors · 2023-11-20
>
> Thank you very much for your time and feedback! We will address the raised questions below.
>
> -) Important baselines are missing. The experiments are more like ablation studies: The authors investigate the performance of GAT/GCN-based solvers with or without the proposed graph sparsification techniques, but does not compare the performance with other solvers mentioned in the related works. To make the contributions strong enough and the results convincing, the paper should compare with the latest methods and outperform them.
>
> The goal of the paper was to show that GNNs can generally increase their performance when learning TSP by applying sparsification. The aim of the paper was not to create one new framework with improved performance but introduce sparsification as an additional “preprocessing” step generally improving the performance of learning-based TSP-solvers using TSP – independent of learning paradigm or the way learning was incorporated in the overall approach (see Encoder-decoder based approaches, search-based approaches and improvement-based approaches in the related work section).
>
> -) The examples in Figure 1 do not make sense. Message passing in GNN does not propagate the features directly, but with a projection matrix (e.g. GraphSAGE). Furthermore, the problem of over-smoothing of GNN not only exist in complete graphs but also in general graphs [1]. How the proposed graph sparsification technique relieve the problem should be more clearly discussed.
>
> We omitted the projection matrix in the message passing process in the figure to make our point clear – The feature vectors get flooded with the information of all nodes in the graph.
> This makes it hard for the decoder architecture to discriminate between the different feature vectors. On the other hand, on the sparsified graph, this problem does not occur.
>
> -) The covered problems only include the 2D tsp, which limits the the contributions of the proposed techniques.
>
> We note that sparsification could be achieved by other sparsification heuristics on higher dimensional data as well. E.g., k-NN is directly generalizable to higher-dimensional data.
>
> -) It lacks necessary theoretical analysis.
>
> Our paper shows the importance of sparsification based on empirical results.
>
> -) The proposed 1-tree sparsification method is derivated from LKH which is a very strong TSP solver. Then the use of the technique in data preprocessing indeed brings prior knowledge to the neural solver. It is very hard to say that whether the better performance comes from the graph sparsification, or comes from the prior knowledge for TSP solving.
>
> We emphasize that we do not run the entire LKH algorithm prior to applying the GNN. Instead, we only use the candidate set generation of LKH based on 1-Trees. With this candidate set, the LKH algorithm would still perform an extensive and time-consuming k-opt heuristic/search which we do not do here for computing our sparse graphs.
>
> -) The size of instances is not given.
>
> We use TSP instances of size n=100 which is also stated in the paper.

---

### Official Review · Reviewer_CDdM · 2023-10-26

**Soundness:** 2 fair
**Presentation:** 3 good
**Contribution:** 2 fair
**Rating:** 3
**Confidence:** 3

**Summary:**

This paper proposes two data preprocessing methods for solving the TSP with GNNs, i.e., k-nearest neighbors heuristic and 1-Trees, which make the corresponding TSP instances sparse by deleting unpromising edges. Experiments are carried out to determine the better sparsification method and the relationships between different data distributions/training dataset sizes and sparsification parameter k.

**Strengths:**

1.	Sparsification (or pruning, candidate selection, etc.) methods are important for solving the TSP as they can substantially reduce the computational complexity, and is commonly used in learning-based algorithms and heuristic algorithms for the TSP.
2.	The paper is overall well written.

**Weaknesses:**

1.	K-nearest neighbors heuristic is already used for sparsification in the input layer (k=20 for TSP100) of GCN by Joshi et al. (2019), Fu et al. (2021) and Xin et al. (2021b) followed this setting. And the proposed “1-Trees” method is similar to the edge candidate set construction process of the LKH algorithm using the 1-tree structure. Thus, the main contribution of this paper seems to be selecting the proper k of k-nn when the problem size is fixed at 100, and transplanting the 1-tree method of LKH as a data preprocessing procedure for learning-based methods. Therefore, the novelty of this paper is not significant enough.

2.	The problem size is fixed at 100 in the experiments so that the generalization ability of the proposed method over different problem sizes is unclear. I recommend the authors add the following question in section 4: how does the problem size n (amount of cities in one TSP instance) relate to the sparsification parameter k?

3.	Comparative experiments with state-of-the-art TSP algorithms is not provided. It is uncertain whether the “1-Trees” method or changing the hyperparameter k of k-nn in existing methods like Joshi et al. (2019); Fu et al. (2021); Xin et al. (2021b) can enhance the performance of state-of-the-art learning-based TSP algorithms.

**Questions:**

1. Please clarify the novelty of this paper in comparisons with the literature papers.

2. how does the problem size n (amount of cities in one TSP instance) relate to the sparsification parameter k?

3. Comparative experiments with state-of-the-art TSP algorithms is not provided.

---

> ### Author Response · Authors · 2023-11-20
>
> Thank you very much for your time and feedback! We will address the raised questions below.
>
> -) K-nearest neighbors heuristic is already used for sparsification in the input layer (k=20 for TSP100) of GCN by Joshi et al. (2019), Fu et al. (2021) and Xin et al. (2021b) followed this setting. And the proposed “1-Trees” method is similar to the edge candidate set construction process of the LKH algorithm using the 1-tree structure. Thus, the main contribution of this paper seems to be selecting the proper k of k-nn when the problem size is fixed at 100, and transplanting the 1-tree method of LKH as a data preprocessing procedure for learning-based methods. Therefore, the novelty of this paper is not significant enough.
>
> Other papers have only considered k-NN so far for a fixed k. Many papers did, however, not perform any sparsification at all. The purpose of the paper is to indicate the importance of sparsification, allowing GNN encoders to increase their performance. We note that the previously considered k-NN approach is severely limited to the uniform data distribution and fails on non-uniform data (consider a graph with two k+1 big coordinate clusters).
> The 1-Tree method has been successfully used within the candidate set generation of LKH, however, to the best of our knowledge we are the first ones to apply it in this setting where we aim to generate promising sparse TSP graphs for GNNs.
>
> -) The problem size is fixed at 100 in the experiments so that the generalization ability of the proposed method over different problem sizes is unclear. I recommend the authors add the following question in section 4: how does the problem size n (amount of cities in one TSP instance) relate to the sparsification parameter k?
>
> We will consider this question in an updated version of the paper.
>
> -) Comparative experiments with state-of-the-art TSP algorithms is not provided. It is uncertain whether the “1-Trees” method or changing the hyperparameter k of k-nn in existing methods like Joshi et al. (2019); Fu et al. (2021); Xin et al. (2021b) can enhance the performance of state-of-the-art learning-based TSP algorithms.
>
> The goal of the paper was to show that GNNs can generally increase their performance when learning TSP by applying sparsification. The aim of the paper was not to create one new framework with improved performance but introduce sparsification as an additional “preprocessing” step generally improving the performance of learning-based TSP-solvers using TSP – independent of learning paradigm or the way learning was incorporated in the overall approach (see Encoder-decoder based approaches, search-based approaches and improvement-based approaches in the related work section).

---

### Official Review · Reviewer_oBRi · 2023-10-29

**Soundness:** 2 fair
**Presentation:** 3 good
**Contribution:** 2 fair
**Rating:** 3
**Confidence:** 4

**Summary:**

The authors observe that the sparsed TSP graph with KNN and 1-tree could improve the performance of the GNN-based method and reduce training time. The topic of studying the sparsity of TSP graph is interesting. The observation also seems reasonable. However, the sparsity like KNN has been used by previous work on TSP and VRP. The used 1-tree method was borrowed from LKH. Therefore, I think the contributions are quite marginal.

**Strengths:**

The observations are interesting.

The experiment design is mostly reasonable.

**Weaknesses:**

Quite some related works about neural-based methods for TSP and VRP are missing, especially from TOP AI conferences.

The sparsity like KNN has been used by previous work on TSP and VRP. The used 1-tree method was borrowed from LKH. Therefore, I think the contribution are quite marginal.

**Questions:**

The results of GAT with dense graphs are quite bad, which makes the results less convincing.

---

> ### Author Response · Authors · 2023-11-20
>
> Thank you very much for your time and feedback! We will address the raised questions below.”
>
> -) Quite some related works about neural-based methods for TSP and VRP are missing, especially from TOP AI conferences.
>
> To the best of our knowledge, we included all relevant related work, but we are happy to extend the related work section with additional papers if the reviewer is open to share which papers in particular are missing.
>
> -) The sparsity like KNN has been used by previous work on TSP and VRP. The used 1-tree method was borrowed from LKH.
>
> To the best of our knowledge, no other papers have extensively studied sparsification so far. If papers applied sparsification, it was typically k-NN with a fixed k and only on the uniform data distribution. We point out that k-NN is not promising on non-uniform data (consider a graph with two k+1 big coordinate clusters). To the best of our knowledge, 1-Tree has not been used for sparsification of graphs afterwards passed to GNNs.

---

### Official Review · Reviewer_gtua · 2023-10-30

**Soundness:** 1 poor
**Presentation:** 2 fair
**Contribution:** 1 poor
**Rating:** 3
**Confidence:** 4

**Summary:**

This paper use one-tree for edge elimination for GNN.
The proposed method achieves an up to ×2 performance improvement w.r.t. the optimality gap and a decrease in runtime by 10% during training and validation, when applied to GCNs. For GATs, the improvements in regards of runtime and optimality gap are even bigger when sparsifying the data first.

**Strengths:**

1. one-tree based sparsity saves time.
2. The introduction and related work sections are well-written.

**Weaknesses:**

This paper is very hard to follow.

1. One-tree has been proposed and existed for many year, introducing sparsity to GNN is not a new idea,
see https://arxiv.org/abs/2006.07054

2. weak evaluation, on TSP 100 only.



We employ GNN for TSP with the aspiration of learning promising edges without the need for human-designed heuristics. However, the use of one-tree heuristics already narrows down the edge set. This means the sparsity is largely dependent on human-designed heuristics rather than data-driven ones.

Also, this sparsity is only limited to TSP and is not able to generalize any other problem.

**Questions:**

1. The table is very confusing, why select different training size? The goal is to investigate how sparsity affect GNN, not training size.

2. How to train your GNN, supervised or reinforcement or even unsupervised? How to get the TSP length?
My understanding is that the code is using reinforcement learning framework based on Jin et al.
But in Jin et al. The authors report a 0.16\% on TSP-100. They further study TSP random200, TSP random500 and TSPLIB from 1~1002.
If the paper use the same model, they should evaluate on the same dataset with Jin et al.

3. In the paper ```We summarize that for the GCN, smaller k led to the overall best results, whereas for the GAT there is a tendency for bigger k (but not dense graphs!) to lead to the best results.```, this is more confusing, that means graph sparsifying can be different for different GNN models, then how we decide $k$ when we use a different GNN model?

4. We report up to ×22 improvements for the optimality gap while reducing the runtime by 50\%.  Can you reveal more details about the training and evaluation, how to get these results?



Jin et al. Deep reinforced multi-pointer transformer for the traveling salesman problem

---

> ### Author Response · Authors · 2023-11-20
>
> Thank you very much for your time and feedback! We will address the raised questions below.
>
> -) One-tree has been proposed and existed for many year, introducing sparsity to GNN is not a new idea
>
> 1-Tree is indeed not a new concept, however, it has not been used in the way we propose: to sparsify TSP graphs for GNN application. The linked paper does not extensively investigate sparsification for GNNs when solving the TSP. They focus on the generalization performance of neural architectures when solving TSP. They only use k-NN which is not promising on non-uniform data (consider a graph with two k+1 big coordinate clusters).
>
> -) Weak evaluation of TSP 100 only.
>
> Smaller instances can often be solved easily with state-of-the-art neural solvers which is why disregarded them. We will try to consider bigger instances in an updated version of the paper.
>
> -) We employ GNN for TSP with the aspiration of learning promising edges without the need for human-designed heuristics. However, the use of one-tree heuristics already narrows down the edge set. This means the sparsity is largely dependent on human-designed heuristics rather than data-driven ones.
>
> We show that sparsification leads to better results compared to dense graphs, nevertheless. We believe that using heuristics in the sparsification process is not necessarily bad.
>
> -) Also, this sparsity is only limited to TSP and is not able to generalize any other problem.
>
> It can be adapted to other problems by using other heuristics for the sparsification process.
>
> -) The table is very confusing, why select different training size? The goal is to investigate how sparsity affect GNN, not training size.
>
> We want to investigate the influence of sparsification on different data set sizes. On bigger datasets, the GNN is (possibly) able to learn the importance of the edge by itself by having access to more data to learn from. On smaller datasets, sparsification is could be more relevant as there is not enough data to learn the importance of all the individual edges.
>
> -) How to train your GNN, supervised or reinforcement or even unsupervised? How to get the TSP length? My understanding is that the code is using reinforcement learning framework based on Jin et al. But in Jin et al. The authors report a 0.16% on TSP-100. They further study TSP random200, TSP random500 and TSPLIB from 1~1002. If the paper use the same model, they should evaluate on the same dataset with Jin et al.
>
> We use a RL framework but point out that our sparsification can be used independently of the used learning paradigm. There are other works that explain how GNNs can be incorporated to capture the different TSP lengths. We did not use the model of Jin et al. directly but used the code as a basis to incorporate our GNN operating on sparse TSP graphs.
>
> -) In the paper “We summarize that for the GCN, smaller k led to the overall best results, whereas for the GAT there is a tendency for bigger k (but not dense graphs!) to lead to the best results.”, this is more confusing, that means graph sparsifying can be different for different GNN models, then how we decide  when we use a different GNN model?
>
> We are indeed unable to derive general rules on how to generally choose the level of sparsification dependent on the chosen GNN. We note, however, that sparsification improves performance and consider the best sparsification level as an additional hyperparameter.